# Evolution and universality of two-stage Kondo effect in single manganese phthalocyanine molecule transistors

Xiao Guo[1,2,3], Qiuhao Zhu[2,3], Liyan Zhou[2,3], Wei Yu[2,3], Wengang Lu[2,3] & Wenjie Liang [1,2,3]✉

The Kondo effect offers an important paradigm to understand strongly correlated many-body physics. Although under intensive study, some of the important properties of the Kondo effect, in systems where both itinerant coupling and localized coupling play significant roles, are still elusive. Here we report the evolution and universality of the two-stage Kondo effect, the simplest form where both couplings are important using single molecule transistor devices incorporating Manganese phthalocyanine molecules. The Kondo temperature $T^*$ of the two-stage Kondo effect evolves linearly against effective interaction of involved two spins. Observed Kondo resonance shows universal quadratic dependence with all adjustable parameters: temperature, magnetic field and biased voltages. The difference in none-equilibrium conductance of two-stage Kondo effect to spin 1/2 Kondo effect is also identified. Messages learned in this study fill in directive experimental evidence of the evolution of two-stage Kondo resonance near a quantum phase transition point, and help in understanding sophisticated molecular electron spectroscopy in a strong correlation regime.

[1] Songshan Lake Materials Laboratory, Dongguan, Guangdong, China. [2] Beijing National center for Condensed Matter Physics, Beijing Key Laboratory for Nanomaterials and Nanodevices, Institute of Physics, Chinese Academy of Sciences, Beijing, P.R. China. [3] CAS Center of Excellence in Topological Quantum Computation and School of Physical Sciences, University of Chinese Academy of Sciences, Beijing, P.R. China. ✉email: wjliang@iphy.ac.cn

Strong correlated many-body effects are among the most intriguing topics in modern solid-state physics, crucial for understanding the fundamental mechanism of exotic phenomena such as non-Fermi liquid behaviors of heavy fermion, the formation of Mott insulator, unconventional super-conductivity, and Kondo physics[1,2]. Although origins of many correlation effects of complex materials are quite different and in the debate, the Kondo effect has risen to be a powerful paradigm to understand strong correlations processes in many systems involving Fermion[1,3] or Bosonic baths[4,5], and impurities with different forms of degenerate, like spin[6–8], charge[9], orbital[10], and topological[11].

Due to the advance in constructing confined nanostructures[6,12], the single-level single-channel spin-1/2 Kondo effect has been widely studied[7,13,14], where a single spin-1/2 local magnetic moment interacts with a large number of itinerant electrons via a single Kondo channel. Kondo phenomena in real materials could be more complicated[15] and more fascinating when more levels and channels involved. In past years, underscreened, overscreened, and multilevel multichannel Kondo phenomena have been intensively studied[9,16,17]. Exotic quasiparticle statistics and quantum phase transition were found in the researches[9,18]. Due to the difficulties in constructing good multilevel Kondo systems[19], and in defining multiple parameters for tuning the systems[20], controlling, and systematic experimental studies of multilevel multichannel Kondo effect are very limited.

Two-stage Kondo effect[13,15,19,21], describing Kondo correlation with two quantum levels via one or two Kondo channels, is the simplest multilevel multichannel Kondo phenomenon. Interaction of particles in two levels competes with the interaction between these localized particles with itinerant ones, leading to nonmonotonic temperature dependence, governed by two Kondo temperatures, $T_k$ and $T^*$. It was observed in artificial quantum dots[19,21], carbon nanotube[22], and molecules[23]. Although the general picture of the two-stage Kondo effect is well-studied, the experimental examination of how Kondo temperatures are related to inner-level interaction is missing, which is essential for understanding the exotic Kondo effect. Moreover, the universality of Kondo physics, an important concept promising wide application of Kondo picture in strong correlate many-body physics, was usually shown as universal scaling of temperature over Kondo temperature[7]. In other cases, the biased voltage was included[14]. As a universal law, universality should be shown against any type of small energy excitations. Here, we report the evolution and universality of two-stage Kondo effects in single Manganese phthalocyanine (MnPC, insert of Fig. 1a) transistors. In this system, the correlation between itinerant electrons and localized spins competes with the correlation between localized spins, leading to the single-channel two-stage Kondo effect. An unambiguously linear relationship was revealed for the first time between $T^*$ and the effective interaction of two spins. The universality of the two-stage Kondo effect is shown against all experimental parameters with a universal quadratic function. Direct comparison between single-channel two-stage Kondo effect and single-channel spin-1/2 Kondo effect in the same device is carried out. Differences in nonequilibrium transport prove that the observed second stage in the two-stage Kondo process cannot be simply regarded as an inverted process of spin-1/2 Kondo effect.

## Results and discussion

**Device fabrication and measurement**. Our devices were prepared by electromigration techniques[12] (details in Supplementary Methods) and a schematic diagram is depicted in Fig. 1a. Figure 1b shows the scanning electron microscopy (SEM) image of a represented junction. The central part of the device is an individual manganese phthalocyanine (MnPc) molecule. Transition metal phthalocyanine molecules benefit from high coupling to metal substrate[24], and their near degenerate molecular orbitals (HOMOs) mainly occupied by d orbital electrons of the metal ion[25]. By changing the transition metal ion, the orbital structure and spin ground state could be easily changed[26], making them ideal for studying exotic Kondo phenomena[20].

Electrical measurement was taken from 280 mK to 20 K, above which the device is less stable. A bias voltage $V_{sd}$ between source and drain electrodes was applied to adjust the energy of electrons passing through the device. While a gate voltage $V_g$ was applied to an aluminum electrode to change the chemical potential of the molecule. The excited states arising from the internal vibration of the molecules help us to identify whether the measured signals come from individual MnPc molecules (see Supplementary Methods).

**Two-stage Kondo effect**. Figure 1c, d shows differential conductance (d$I$/d$V$) plots of two representative molecular devices (1 and 2), as a function of bias voltages ($V_{sd}$) and gate voltages ($V_g$). Typical coulomb blockade behaviors were observed in both devices, with differential conductance peaks defining two-electron blocking regions, I and II, associated with two adjacent charge states of the molecule.

A sharp zero-bias peak was observed in the region I while a gate tunable zero-bias split peak was observed in region II for both devices. The evolution of these peaks around zero bias forms the central topics of this work, and as we will see they are different Kondo processes corresponding to different conditions in different charge states of MnPc molecules. For simplicity, further discussions are mainly based on device 1, although all the features are shared by devices 1 and 2.

Temperature, electrical, and magnetic study of the zero-bias conductance peak in region I proves single-channel spin-1/2 Kondo resonance, with a single Kondo temperature $T_{k,1/2} = 3.21 \pm 0.02$ K (at $V_g = 2$ V) and a g factor of 1.79 (see Supplementary Methods). In region II, a pronounced sharp dip is observed within a large broad peak (Fig. 2a), forming a split peak feature. The split peak shifts significantly with applied gate voltage (Fig. 3a). The dip becomes shallower and disappears when temperature increases from 0.3 K to 3 K. As the temperature increases further, the broad peak decreases and eventually vanishes. The zero-bias conductance shows a distinct nonmonotonic temperature dependence (Fig. 2b). These features, combining together, demonstrate a clear two-stage Kondo effect in single MnPc transistors[13].

The spin state in region II could be either spin singlet or triplet with the two-electron occupation. The two-stage Kondo effect can appear in both cases[13,15]. A magnetic field was applied to determine the ground state of the device in this region (Fig. 2d, e). The crossing and linear dependence behavior of splitting peaks demonstrate a magnet field-induced ground-state transition from spin singlet (at lower magnet field) to spin triplet (higher magnet field)[23]. The corresponding energy diagram is shown in Fig. 2c. The finite bias peak is associated with excitations into spin triplet[27]. The energy of triplet state is 0.45 meV higher than that of singlet state at $V_g = -2$ V in zero magnetic field. The singlet to triplet (S–T) transition field is 3.2 T. g factor is 1.76, very close to the value obtained in the region I. It is worth noting that the splitting peak feature can also appear when the ground state is triplet if the anisotropy of the molecule is introduced[28]. However, it is not the case for our devices as the transition resonances between spin singlet and triplet are clearly resolved in Fig. 2e.

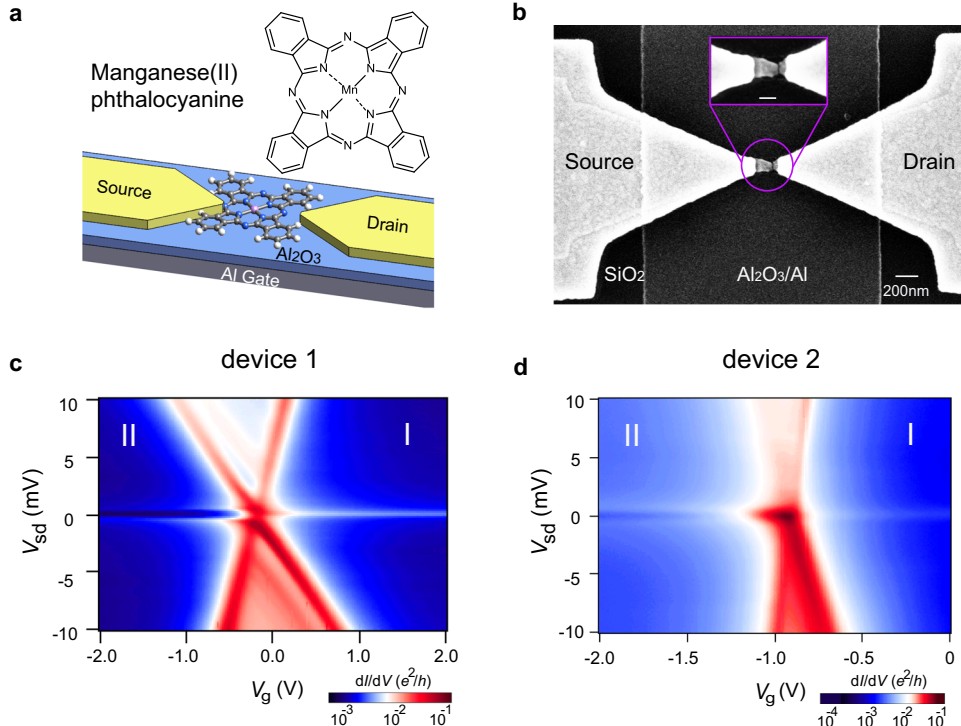

**Fig. 1 Device and electron transport spectra. a** Structure of Manganese(II) phthalocyanine molecule (MnPC, top), a schematic diagram of a single MnPC molecule transistor device (bottom). **b** SEM image of the device after breaking: gold nanowire over an $Al_2O_3$/Al gate, a nanogap formed in gold nanowire by electromigration. The scale bar of the inset is 100 nm. **c, d** Color plots of differential conductance ($dI/dV$) as a function of bias voltage ($V_{sd}$) and gate voltage ($V_g$) for device 1 at 280 mK (**c**) and device 2 at 1.4 K (**d**).

Singlet two-stage Kondo effect is predicted to be a single-channel process near a Kosterlitz–Thouless (KT) type quantum phase transition point and has been confirmed in single $C_{60}$ molecule devices[13,23,29]. Transport in our single molecular devices is close to the quantum phase transition point with only 0.45 meV between singlet and triplet states. No significant enhancement of resonance at the critical field (Fig. 2d and Supplementary Fig. 3) demonstrates transport in our molecular devices is a single-channel process[8,23]. Our scenario is consistent with the proposed theory for the single-channel two-stage Kondo effect[13,29].

**Extraction and evolution of $T^*$.** Observed nonmonotonic singlet two-stage Kondo could be explained by competition between molecular internal binding correlation ($k_B T^*$) and the spin screening correlation by itinerant electrons ($k_B T_k$) via a single Kondo channel. At finite temperature, the spin-singlet state could dissociate into two independent spin due to quantum criticality near the KT quantum phase transition[23]. At higher temperatures, itinerant electrons in the electrodes screen one of the two spins in the molecule, leading to a broad spin-1/2 Kondo peak, characterized by $T_k$[7], (Supplementary Eq. (1)). At lower temperature, the singlet binding energy cause the second spin to destroy the screening cloud of the first stage Kondo resonance, forming a Kondo scattering in the second stage, previously described by an inverted spin-1/2 Kondo resonance form of[22,23]

$$G(T) = G_0 - G_0 / \left[ 1 + (2^{1/s} - 1)(T/T^*)^2 \right]^s + G_c \quad (1)$$

where $T^*$ is the second Kondo temperature, $G_0$ is a typical conductance value, $G_c$ is the background conductance. $T_k$ and $T^*$ were extracted accordingly. $T_k = 13.3 \pm 2.7$ K and $T^* = 4.1 \pm 0.1$ K are extracted at $V_g = -2$ V, much higher than the previous study in $C_{60}$ molecule quantum dot[23], making it possible for us to study

the evolution of the two-stage Kondo effect at a broader energy range.

Evolution of $T_k$ and $T^*$ were hard to control and understand in previous studies, even in cases where multiple gates were applied to tune the correlation parameters[7,19,21]. In our single MnPc transistors, however, a significant tunability of $T^*$ with a single gate is shown (Fig. 3c and Supplementary Fig. 4). Extracted $T^*$ increases linearly by a factor of 2 with gate for device 1. This clear evolution of the two-stage Kondo effect helps us to understand how the Kondo temperatures could be linked to the internal interaction of the molecule devices.

Gate tuning of molecule transistor and quantum phase transition have been demonstrated before[23]. But nature of gate tuning of $T^*$ have never been successfully examined. For spin-1/2 Kondo effect, the single Kondo temperature can be tuned by changing the molecule's chemical potential via a gate voltage, since it is linked to coupling strength of local impurity to itinerant electrons. The gate tuning of $T^*$ in the two-stage Kondo effect was unexpected since it represents the internal singlet binding energy of two electrons in the molecules and may not be affected by external gate voltage[7].

The reason for this unexpected behavior could arise from a gate tunable singlet–triplet level spacing in our devices. Figure. 3a, b shows that the energy difference between singlet and triplet ($\Delta\varepsilon = \varepsilon_T - \varepsilon_S$ the singlet–triplet energy difference, $\varepsilon_S$ is spin-singlet energy, $\varepsilon_T$ is spin triplet energy) become smaller at more negative gate voltages, while the opposite trend is observed in device 2 (Supplementary Fig. 5). This gate induced changes in singlet–triplet splitting, as has been observed in carbon nanotube[30] and $C_{60}$ molecular device[23], could be due to different gate couplings of the two orbitals, or level renormalization effect induced by asymmetric tunneling couplings[30]. The different gate dependence of device 1 and 2 might come from different local coordination to the electrodes.

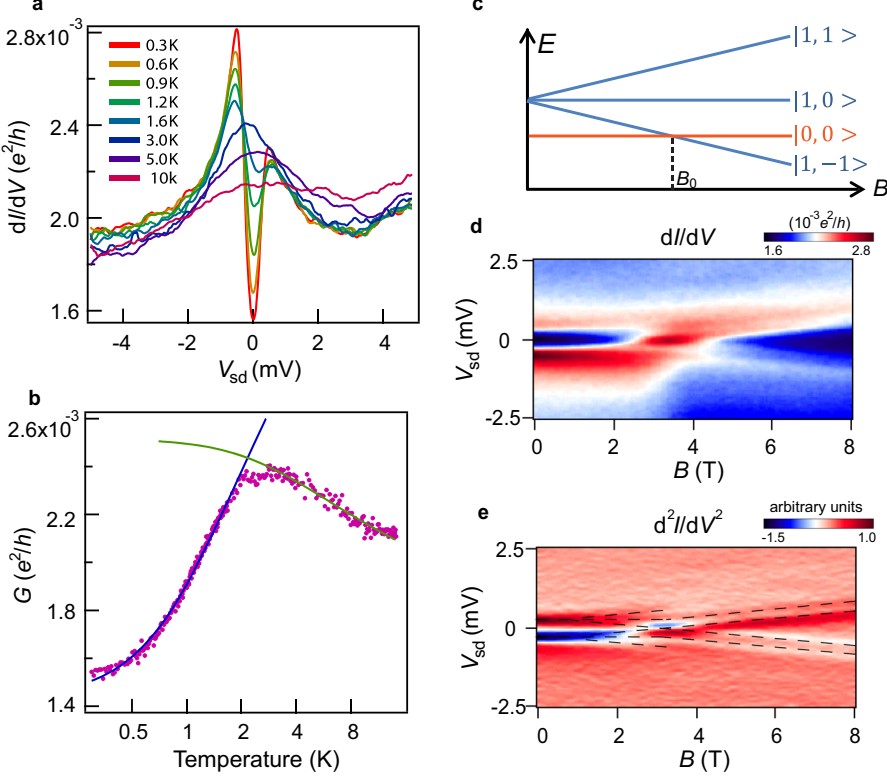

**Fig. 2 Two-stage Kondo effect. a** Plot of differential conductance d$I$/d$V$ against $V_{sd}$ at $V_g = -2$ V (region II) of device 1 at various temperatures. **b** Temperature dependence of the zero-bias conductance $G(T)$ at $V_g = -2$ V. The green and blue lines are fitting results to spin-1/2 fully screened Kondo model (Supplementary Eq. (1)) and inverted spin-1/2 Kondo model (Eq. (1)), respectively. **c, d** Differential conductance d$I$/d$V$ as a function of $B$ and $V_{sd}$ at $V_g = -2$ V and temperature $T = 280$ mK (**d**). Energy level sketch of magnet induced singlet to triplet transition process (**c**). $B_O = 3.2$ T is the critical point where the transition occurs. **e** Numerical derivative of d$I$/d$V$ in **d**. The black dash line indicates the peak position, which corresponds to the transition lines between spin singlet and triplet.

The evolution of $T^*$ vs. $\Delta\epsilon$ is plotted in Fig. 3c, d, where $\Delta\epsilon$ is acquired by taking the peak position of the split peaks at different gate potentials. A clear linear relation is established in Fig. 3d and the line intersect $\Delta\epsilon$ axis at ~0.35 meV. It was theoretically proposed that $T^*$ in the two-stage Kondo process would decrease when approaching S–T transition point, as a function of effective exchange interaction $\Delta I$ between the two impurities ($\Delta I = \Delta\epsilon + J/4$, $J$ is the exchange energy between the local electrons)[13]. $T^*$ was predicted to follow a linear dependence with $\Delta I$, when $\Delta I$ is comparable to $T_k$, but follows an exponentially decay $\exp[-T_k/\Delta I]$ when $\Delta I$ is much smaller than $T_k$. In device 1, a clear linear dependence is established when obtained $\Delta\epsilon$ is on the order of 0.5 meV while $k_B T_k$ is on the order of 0.9 meV, comparable to each other. Assuming $T^* = k\,\Delta I$, i.e., $T^* = k(\Delta\epsilon + J/4)/k_B$, linear fit in Fig. 3d shows $k = 0.30 \pm 0.02$ and $J = -(1.40 \pm 0.04)$ meV. The exchange energy of $-1.4$ meV reveals a ferromagnetic interaction between the two electrons in our MnPc molecular devices. It is obvious the nature and magnitude of electron exchange energy in the molecule system would change the evolution of $T^*$ as a function of $\Delta\epsilon$. Similar analysis could be applied to device 2 (Supplementary Fig. 5). A value of $J = -1.6$ meV could be deduced, quantitatively in agreement with device 1. The close similarity of $J$ from two separated devices with different local environments suggests $J$ is an internal parameter of the molecule and merit of the current device structure. Considering the ferromagnetic interaction and almost degenerate spin single and triplet, the two levels contributing to the two-stage Kondo effect in our devices are most likely to be $3d_{xz}$ and $3d_{yz}$ of the Mn atom in MnPc

molecule[31]. In molecular devices, both positive and negative $J$ can lead to a singlet ground state as long as $\Delta I$ is positive. Indeed, antiferromagnetic coupling in the different molecule could be inferred from previous single-molecule device study[23] following the same analysis.

**Universality vs. temperature, voltage, and magnetic field**. The universality of the Kondo effect, independent of physical systems and origin of the interactions, reveals the great impact of the Kondo theory and wide application of the experimental results. Most previous works only address Kondo resonance's universality in a single-level system with limited parameters. We here examine the universality of single-channel two-stage Kondo effect by thermal, electrical, and magnetic excitations. We address their similarity and difference with the single-channel spin-1/2 Kondo effect in an identical device.

Close to zero temperature, in the second stage, the low-energy excitation behavior of the two-stage Kondo effect is dominant by the smaller energy scale $T^*$. In Fig. 4a, the zero-bias normalized conductance $(G - G_c)/G_0$ at different $V_g$ are plotted as a function of $T/T^*$. The equilibrium conductance (conductance at zero bias) of the two-stage Kondo effect shows a clear invert scaling form of the spin-1/2 Kondo effect, similar to previous observations in carbon nanotube[22] and singe $C_{60}$ molecule devices[23]. In addition, the nonequilibrium conductance (conductance at finite bias) at different gate voltage can also be scaled to an identical dip profile when the bias is scaled by $T^*$, as shown in Fig. 4b. Comparing Fig. 4a, b, the resonance profile of the two-stage Kondo effect tends to deviate from universal function at a lower energy ($T/T^*$

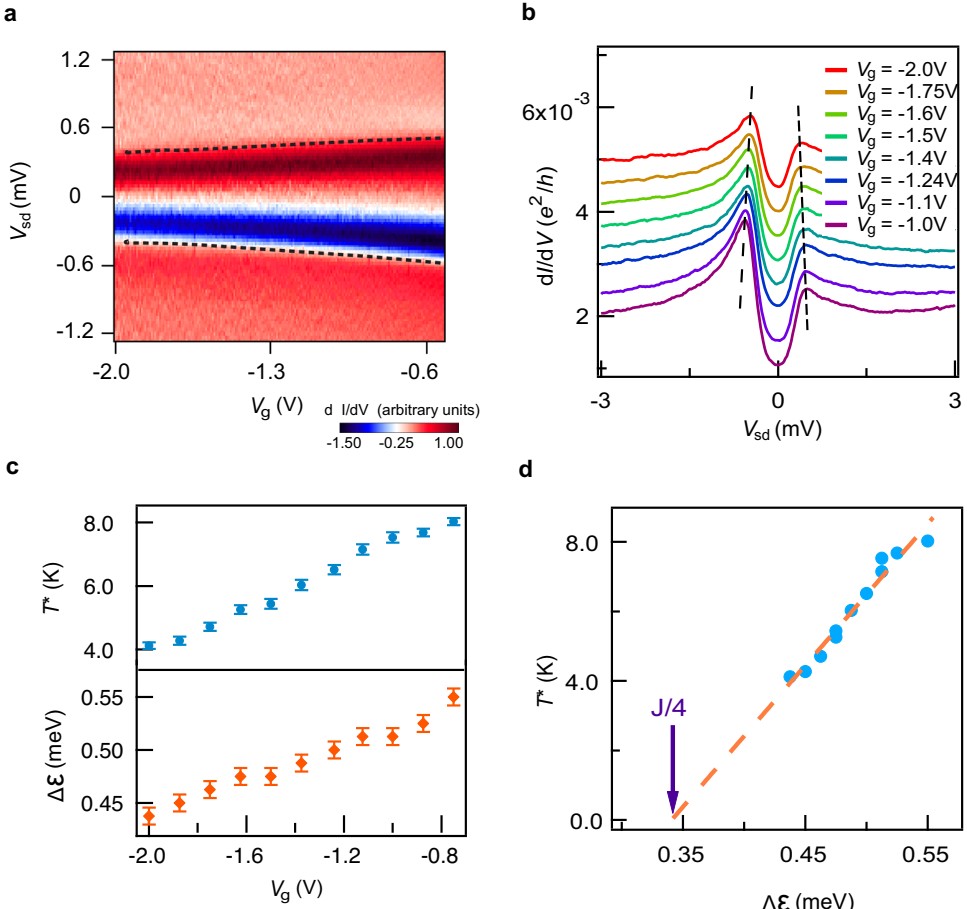

**Fig. 3 Gate voltage dependences of $T^*$ and singlet–triplet energy splitting. a** Color plot of the numerical derivative of d$I$/d$V$ as a function $V_{sd}$ and $V_g$ in region II for device 1. The black dash line indicates the peak positions, which corresponds to the energy difference between spin singlet and triplet. **b** Line traces of d$I$/d$V$ against $V_{sd}$ in a at different gate voltages. The peak positions are marked by a dash line. The curves are shifted vertically for clarification. **c** $T^*$ and $\Delta\varepsilon$ of device 1 at different gate voltage. The error bars of $T^*$ represent the uncertainty of the fitting result using Eq. (1). $\Delta\varepsilon$ is acquired by taking the peak position of the split peaks at different gate potentials in **b** and the uncertainty is also represented by error bars. **d** Plot of $T^*$ against $\Delta\varepsilon$ for device 1. The dash line is a linear fitting.

or $V_{sd}/T^*$) when $V_g$ becomes more close to the charge degenerate point between region I and II. This can be explained by the higher $T_k$ near the charge degenerate point affecting the nonequilibrium Kondo effect more strongly, manifesting again the competition between itinerant correlation and impurity correlations.

In the low-energy regime ($k_BT$, $eV$ and $gu_BB << T_{kondo}$, where $T_{kondo}$ represents $T_{k,1/2}$ or $T^*$ for spin-1/2 Kondo effect or two-stage Kondo effect), the universality of the Kondo effect should also manifest as a strikingly similar response upon different kinds of perturbations as long as the based dynamics are similar[14]. For the two-stage Kondo effect, the conductance suppression in the second stage would weaken upon excitation by temperate ($T$), electrical voltage ($V$), and magnet field ($B$) as they all contribute to quench molecular internal binding energy. Our devices indeed show similar power-law dependence[32] with temperature ($T$), bias voltage ($V_{sd}$), and magnet field ($B$) (Supplementary Fig. 6). To determine the scaling relationship for $T$, $V_{sd}$, and $B$, we fit the low-energy conductance to the form (Supplementary Fig. 6),

$$G(V,T,B) = G_0 \left[ C_V \left( \frac{eV_{sd}}{k_B T^*} \right)^{P_V} + C_T \left( \frac{\pi T}{T^*} \right)^{P_T} + C_B \left( \frac{gu_B B}{k_B T^*} \right)^{P_B} \right] + G_c$$

(2)

where $P_v$, $P_T$, $P_B$ are scaling exponents and $C_v$, $C_T$, $C_B$ are the scaling coefficients[14,33]. The best fit exponents are $P_V = 1.98 \pm$

0.08 $P_T = 1.83 \pm 0.01$ $P_B = 1.89 \pm 0.09$, closing to 2, and revealing a Fermi liquid behavior[33]. Treating all the exponents to be 2, We plot the low-energy excitations against $\left( \frac{eV_{sd}}{k_B T^*} \right)^2$, $\left( \frac{gu_B B}{k_B T^*} \right)^2$ and $\left( \frac{\pi T}{T^*} \right)^2$ represented by a unitless value $X^2$ in Fig. 4c, in which the same quadratic behaviors are valid up to $X^2 = 0.5$. This valid quadratic dependence range is substantially large[14], considering it should be only valid for the second-order approximation in Fermi liquid theory. A similar analysis can be applied to the spin-1/2 Kondo effect in our devices (see Supplementary Fig. 7) and the low-energy conductance show also quadratic power relation with temperature and bias voltage, consisting with previous theorical[33] and experimental studies[14,34].

The coefficients in Eq. (2) ($C_v$, $C_T$, $C_B$) are not necessary to be identical. Multichannel multilevel Kondo theory predicted their values and the ratio between them is determined by the interplay between different levels and their coupling strength to conduction channels. Our fitting gives $C_v = 0.36$, $C_T = 0.27$, and $C_B = 0.37$, with $C_v/C_B = 1$ and $C_T/C_B = 0.7$. As a comparison, recent prediction for multichannel multilevel Kondo effect[34] proposes $C_v/C_B = 1.5 + 9F$ and $C_T/C_B = 1 + 3F$, where $F$ is Fermi liquid constant, describing relative strength of orbital–orbital coupling to orbital–channel couplings. Our finding qualitatively supports this prediction with $F \sim -0.06$.

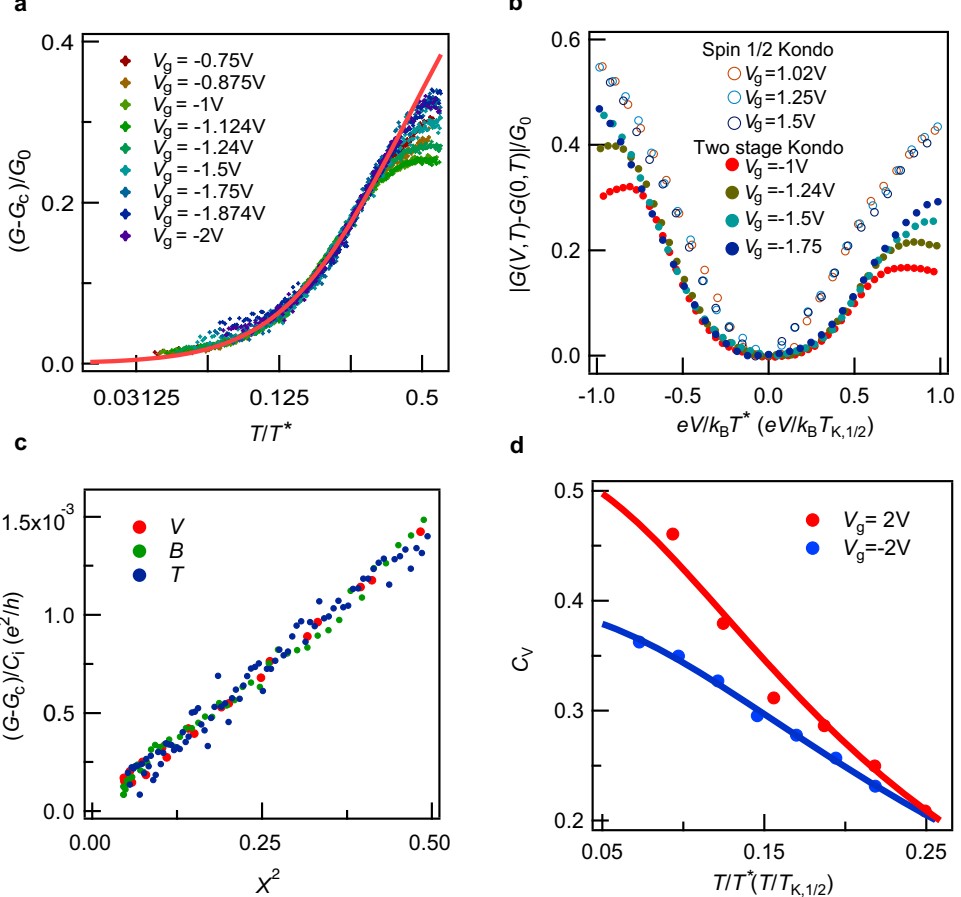

**Fig. 4 Universal scaling of two-stage Kondo resonance. a** The normalized zero-bias conductance, $(G–G_c)/G_0$ for device 1 in region II at various gate voltages, shows a universal function of $T/T^*$. The solid line is the scaling form of invert spin-1/2 Kondo resonance. **b** The differential conductance vs. $V_{sd}$ curves at different $V_g$ are scaled to an identical line by $T^*$ or $T_{k,1/2}$ **c** Plot of scaled low-energy differential conductance $(G–G_c)/C_i$ ($V_g = −2$ V and $C_i$ are the coefficients, $i =$ V, T and B) against $\left(\frac{eV_{sd}}{k_BT^*}\right)^2$, $\left(\frac{\pi T}{T^*}\right)^2$ and $\left(\frac{g\mu_BB}{k_BT^*}\right)^2$ (represented by $X^2$). The linear relation between conductance and $X$ demonstrates a quadratic behavior for $T$, $V$, and $B$. **d** Values of $C_v$ for two-stage Kondo ($V_g = −2$ V) and spin-1/2 Kondo effect ($V_g = 2$ V) at different temperatures. The solid lines are fitting to Eq. (3).

**Nonequilibrium scaling**. Nonequilibrium scaling form of two-stage Kondo effect shows distinct difference to spin-1/2 Kondo effect (Fig. 4b). The nonequilibrium conductance of the two-stage Kondo effect varies much slower with scaled bias than the spin-1/2 Kondo effect. We extract $C_v$ at different temperatures for two-stage and spin-1/2 Kondo effect and plot the result in Fig. 4d. The $C_v$'s of the two-stage Kondo effect are much smaller than that of the spin-1/2 Kondo effect, consistent with the result in Fig. 4b.

The temperature-dependent $C_v$ can be described in a formula as:[14,22,34]

$$C_V = A \frac{\pi^2 C_{T0} \alpha}{1 + \pi^2 C_{T0}\left(\frac{\gamma}{\alpha} - 1\right)\left(\frac{T}{T_{Kondo}}\right)^2} \quad (3)$$

where $A$ is a factor, $T_{Kondo}$ represents $T_{k,1/2}$ or $T^*$, $\alpha$ and $\gamma$ are the scaling coefficients. Here, $C_{T0} = 0.50$ is acquired by expanding Eq. (1) close to zero temperature, as $G(T,0)−G_c \approx C_{T0}G_0(\pi T/T^*)^2$, where $G(T,0)$ represents $G(T)$ at $V_{sd} = 0$. For spin-1/2 Kondo effect, $A = [G(T,0)−G_c]/G_0$, while for the two-stage Kondo, $A = [G_0 + G_c−G(T,0)]/G_0$. The coefficients $\gamma$ and $\alpha$ characterize the temperature broadening and zero-temperature curvature of the Kondo peak (or dip) respectively. When $T \approx 0$, $C_V/C_{T0} = \alpha\pi^2$. The fitting result in Fig. 4d gives $\alpha = 0.11 \pm 0.01$, $\gamma = 0.48 \pm 0.06$ for spin-1/2 Kondo process, and $\alpha = 0.08 \pm 0.01$, $\gamma = 0.23 \pm 0.01$ for the two-stage Kondo effect. The extracted $\alpha$ for spin-1/2 Kondo

effect in our MnPc devices is very close to the previous value ($\alpha = 0.10 \pm 0.015$, $\gamma = 0.5 \pm 0.1$)[14], but the value of $\alpha$ in two-stage Kondo effect is much smaller ($\alpha = 0.13 \pm 0.03$, $\gamma = 0.55 \pm 0.3$)[22].

From the analysis above, although the dip profile of nonequilibrium two-stage effect follows the similar scaling form with spin-1/2 Kondo resonance, the different scaling coefficient indicates that the second stage of the two-stage Kondo effect cannot be simply assumed as the invert process of the spin-1/2 Kondo[22]. Future advances in nonequilibrium theoretical examination into the two-stage Kondo effect could help to elucidate the nature of this discrimination.

Previous STM studies of metal phthalocyanine molecules often result in a Kondo resonance[35,36] when metal tip (electrode) sitting on top of the transition metal ion the molecule. The site-dependent Kondo effect and two-stage Kondo effect were also shown in these studies[20,24]. where a dip often appears when transition metal ion is on top of Au atom, disappears when ion is in interstitials position, suggesting a single-channel event. Our identification of the single-channel two-stage Kondo effect of a single MnPC molecule agrees with these studies, offering a dept understanding of relevant correlation processes. We believe electron transport via our molecule devices is also mainly through Mn ion, although the ground state is 1/2 and 0 (1) for different redox states. The spin ground state of an isolated MnPc molecule is 3/2 with unpaired spin occupying $dz^2$ and degenerate

$d_{xz}$ and $d_{yz}$ orbitals of Mn atom[25]. It's believed a MnPc molecule on Au(111) surface also hold spin 3/2 ground state, backed by DFT calculation[37,38] and existent Kondo peak in STM studies. However, contrary to general thought, spin 3/2 magnetic impurity can only result in an underscreened Kondo process[16], which produces a much weaker Kondo resonance and has a much lower Kondo temperature comparing with spin-1/2 magnetic impurity. Such an underscreened Kondo process has never been discovered in previous STM studies for molecular magnet[35,36,39]. It cannot rule out the possibility ground state of MnPc on Au might occupy other ground states. DFT calculations indicate stronger interaction of molecular magnet with Au substrate lead to weaker magnetic property which is often the case of metal phthalocyanine molecules[37,38]. Furthermore, the spin state of MnPc molecules usually alters when laying on metal substrates[40] or with coordination of small molecules, like CO[35], H atom[36] even Li atom[41], where changes in chemical bonds or ligand coordination lead to the redistribution of the electron in the 3d orbitals. The current-induced breaking technique usually creates molecular junctions with rather complex local coordination of molecule-Au electrode interface. That may reduce the spin state of the molecule in our devices from 3/2 to 1/2.

In conclusion, we studied gate evolution and universality of the two-stage Kondo effect using a single-molecule transistor platform. We experimentally established the dependence of second stage Kondo temperature $T^*$ to the effective exchange energy of two molecular levels, explained the origin of gate tunable $T^*$. We identified ferromagnetic exchange coupling between electrons in MnPc devices. The universality of two-stage Kondo resonance against all measurable parameters is determined for the first time, the result agrees well with recent prediction in multilevel multichannel Kondo theory. The difference between single-channel two-stage Kondo effect and single-channel spin-1/2 Kondo effect is attributed to the difference in $C_v$, the universal coefficient for out of equilibrium electrical excitation. The study and manipulation of the two-stage Kondo effect, using a single-molecule transistor, demonstrate a powerful platform for quantitatively study sophisticate strong correlation processes. Deliberate choosing kinds of molecules and electrode couplings will lead to further fruitful studies of multichannel multilevel correlation and potentially benefit in multiple fields.

## Data availability

The data that support the findings of this study are available from the corresponding author on reasonable request.

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

## Acknowledgements

We thank Professor Yifeng Yang, Lu Yu, David Goldhaber-Gordon, Jan von Delft, Honghao Tu, Yuyang Zhang, and Tao Xiang for helpful discussions. This research is supported by the National Basic Research Program of China (2016YFA0200800), Strategic Priority Research Program of Chinese Academy of Sciences (Grant No. XDB30000000), Strategic Priority Research Program of Chinese Academy of Sciences (Grant No. XDB07030100), Sinopec Innovation Scheme (A-527).

## Author contributions

X.G. and W. Liang conducted the experimental design. X.G. made a major contribution in device fabrication and transport measurements. Q.Z., L.Z., and W.Y. helped in device fabrication and transport measurements. W. Liang, X.G., Q.Z., and W. Lu. analyzed the data and wrote the paper. All the authors discussed the results and commented on the paper.

## Competing interests

The authors declare no competing interests.
