## [Peer Review File · Nature Communications]

Reviewers' comments:

Reviewer #1 (Remarks to the Author):

In this manuscript, Guo et al report the two-stage Kondo effect in single molecule transistors containing a magnetic MnPc molecule. They fabricated the transistor and measured the transport properties systematically as functions of gate voltage (V_g), temperature (T) and external magnetic field (B). They find different Kondo signatures in region I and II as a function of V_g . In the former, typical $S=1/2$ Kondo effect occurs and the singlet-triplet Kondo effect occurs in the latter region. The results are essentially the same as those reported previously in Phys. Rev. Lett. 2002 and Nature 2008. However, the universal scaling as functions of bias voltage, T and B found in the present study look originally new. The scaling of typical $S=1/2$ Kondo effect is well established, but those relevant to multiple spins, orbitals and channels remain to be explored widely. Thus, the results of Guo et al would impact not only to Kondo physics but also to the many-body correlation physics and non-equilibrium transport physics. However, this reviewer is not satisfied with the current presentation.

(1) The electronic/spin configurations of MnPc in the devices and the relation with the Kondo screening are quite elusive. Give more clear views on in the regions I and II with molecular orbital diagram and experimental/theoretical results about MnPc state inside the device. This is a serious problem common to the single molecule junction studies in which a molecule bridges two electrodes. More than twenty years have passed since the first report on the Kondo effect in nano-structured devices. It is disappointed that the central molecule in the device remains ill-defined and no progress is made about the identification. The scanning tunneling microscopy studies of two stage Kondo effect cited in this manuscript provide more clear relations of spectral Kondo signatures with the electronic/spin configurations and also the geometrical structure.

(2) The authors discuss the molecular state by comparing the adsorption of MnPc on flat Au(111) surface. When assuming the similar structure in the device, I am wondering the molecule can be gated. MnPc on Au(111) has been well investigated. Cite the papers such as Phys. Rev. B 84, 125446 (2011) and J. Phys. Chem. C 112, 13650 (2008) and compare the present results with the previous ones.

(3) The vibrational signatures at 16 and 63 mV are presented as solid evidence of a single molecule device in FIG.S1. The inelastic structures should appear symmetrically in the I-Vsd spectrum. Show more clear spectra like typical cross-sections of 2D color map.

Reviewer #2 (Remarks to the Author):

The authors present in their work the study of two-stage Kondo effect using a single MnPc single molecule transistor. They perform transport measurements as a function of temperature, gate voltage and magnetic field in the two regions of the Coulomb diamond to discriminate the spin $1/2$ to the two-stage Kondo effect. They then perform measurements to show the gate control and the universality of the two stage Kondo effect. The results are important, however, I cannot recommend this work for publication in Nature communications at this stage, as I do not clearly understand the novelty compared to the results obtained by N. Roch et al. Nature Nature 453, 633 (2008) (Quantum phase transition in a single-molecule quantum dot), except the careful study of the evolution of T^* as a function of the singlet-triplet energy. If the authors clearly state the novelty of their work, I will of course change my opinion.

I also have different comments :

1- The authors present in the SI the temperature study of the zero-bias anomaly in region 1. As a saturation of the conductance is clearly observed at low temperature, the authors assume (with a comparison to theory) that a spin 1/2 Kondo effect is observed on the right, with a Kondo temperature of the order of 3.21K. The authors also extracted the g factor performing measurements as a function of the external magnetic field. My comment is : what is the Kondo temperature obtained from the measurement as a function of the magnetic field ? Is it consistent with the TK obtained as a function of temperature ?

2- In Figure 1.c, the authors observe a clear dip in region 2. Measurements as a function of temperature (Figure 2) are consistent with a two-stage Kondo effect. However, in Figure 2.c, it is absolutely not clear that the ground state is the singlet, as the clear signature of the triplet state (3 transitions) should be observed before the critical field, as observed in the work of N. Roch et al., but also in the work of Yu. Bomze et al. Phys. Rev. B 82, 161411(R) (2010) (Two-stage Kondo effect and Kondo-box level spectroscopy in a carbon nanotube). After the critical field, 2 transitions should also be observed. Could the authors comment on why it is not observed ? Or maybe it is just not visible from the figure which is presented, the authors could then apply a second derivative of the conductance to observe this signature.

3- In Figure 3.a, the authors present the gate voltage dependence of the singlet-triplet energy splitting. It looks like for more negative gate voltages, it could have been possible to observe the underscreen spin 1 Kondo effect. I presume that the authors did not perform measurements for lower gate voltages to prevent leaking currents or breaking of the Al gate ?

4- Could the authors comment the influence of the magnetic properties of MnPc of the Kondo signatures, as it is not clear for me (I apologize if I missed this point) if transport occurs through the Pc ligand or the Mn atom. My guess is through the Pc ligand as the evolution of the conductance as a function of temperature in region 1 does not fit with NRG calculations for $S=3/2$.

Response to referees

Dear referees:

Before we respond to questions raised by referees, we want to thank referees for helpful and constructive critics and suggestions. A major revision is made in this manuscript. The title is changed to "Evolution and universality of Two-stage Kondo effect in single manganese phthalocynine molecule transistors". The main focus of this manuscript emphasizes on experimental study on evolution of two stage Kondo effect against inner and outer degree of freedoms exhibited by our molecule devices and examine some of important theoretical predictions. In the first half of this manuscript, after describing how we construct the device and demonstrate establishment of two stage Kondo system, we continue to analysis the nature of gate tunable T^* , which should only be affected by inner degree of freedom. We found it could be explained by gate dependent effective exchange coupling strength ΔI . A long predicted linear dependence of T^* to ΔI is revealed for the first time. In the second half of this manuscript, we examined the universality of two stage Kondo effect upon external excitations. Unlike other works discussed universality of Kondo effect, our data offer opportunity to examine universality of two stage Kondo effect vs all experimental adjustable parameters: temperature, voltage and magnetic field. Although the detailed mechanism of how these physical processes change Kondo resonance vary, they showed an impressive similarity (quadratic relation) after scaled with energy $k_B T^*$, agreed to Fermi liquid behavior. More technique details to strengthen the analysis and conclusions are added. The discussion about the configuration of the molecule in device is left out as we realize than the assumed configuration in last manuscript may not be the only configuration exhibiting Kondo effect. Besides, we add new analysis and discussion about comparison between two kinds of Kondo effect, revealing the second stage of two-stage Kondo effect is not simply the invert process of spin 1/2 Kondo effect. In the following, we will address referees' question one by one. Thank you very much in helping us revising our work!

Reviewers' comments:

Reviewer #1 (Remarks to the Author):

In this manuscript, Guo et al report the two-stage Kondo effect in single molecule transistors containing a magnetic MnPc molecule. They fabricated the transistor and measured the transport properties systematically as functions of gate voltage (V_g), temperature (T) and external magnetic field (B). They find different Kondo signatures in region I and II as a function of V_g . In the former, typical $S=1/2$ Kondo effect occurs and the singlet-triplet Kondo effect occurs in the latter region. The results are essentially the same as those reported previously in Phys. Rev. Lett. 2002 and Nature 2008. However, the universal scaling as functions of bias voltage, T and B found in the present study look originally new. The scaling of typical $S=1/2$ Kondo effect is well established, but those relevant to multiple spins, orbitals and channels remain to be explored widely. Thus, the results of Guo et al would impact not only to Kondo physics but also to the many-body

correlation physics and non-equilibrium transport physics. However, this reviewer is not satisfied with the current presentation.

Comment: Current work is partially in-line with Nature 2008 work in that both work note the existence of two stage Kondo effect in molecular devices but go further in difference direction: Nature 2008 work went on discussing the fingerprint of quantum phase transition while current work emphasis on behavior of two stage Kondo effect including evolution of Kondo temperature in case of two stage Kondo effect under influence of gate and full spectrum quadratic universality of two stage Kondo after scaled with Kondo temperature which have never been done in past twenty or thirty years. As also been pointed out by the referee in the first half of current research our finding is consistent with the PRL 2002 paper while the 2002 paper is only a theoretical prediction that has stood for verification till today. Our experimental finding proves result of their calculation based on numerical renormalization group theory is correct. We believe these are exciting progress. We do agree with the referee that with demonstration of universality of complex Kondo effect, our work would impact the correlation physics and non-equilibrium physics wider than conventional Kondo physics treating only spins or single level impurities. Thanks for your opinion and support.

Q(1) The electronic/spin configurations of MnPc in the devices and the relation with the Kondo screening are quite elusive. Give more clear views on in the regions I and II with molecular orbital diagram and experimental/theoretical results about MnPc state inside the device. This is a serious problem common to the single molecule junction studies in which a molecule bridges two electrodes. More than twenty years have passed since the first report on the Kondo effect in nano-structured devices. It is disappointed that the central molecule in the device remains ill-defined and no progress is made about the identification. The scanning tunneling microscopy studies of two stage Kondo effect cited in this manuscript provide more clear relations of spectral Kondo signatures with the electronic/spin configurations and also the geometrical structure.

A1 Identification the geometrical structure as well as the molecular orbital configurations have long been a challenge task in the studies of solid-state single molecule devices. Although great effort has been devoted to, we are sorry that we can't solve this problem at this stage. The adopted break junction techniques offers great advantages in studying transport spectra, with most stable electronic structure, finest energy resolution, great gate control, giving detailed information of energy spectra and transport process, with the sacrifice of losing geometry information. Effort in constructing the structural imaging at same time of extracting the sophisticated transport information is beyond current work. We agree with the referee that having both would be ideal but current technique of STM and three terminal devices as showed in this work was non-compatible. As referee can see, we study and learn much valuable information from STM experimental works. But at the same time we do argue that spectrum-wise, STM is still not as good as molecular transistor structure. It's hard to resolve energy scale close and less than milli electron volt. It can't control and adjust temperature of the molecule and STM tip precisely. It can't offer a third knob to tune the local chemical potential of the molecule to clarify the spectrum. And with all the effort put in, it's still hard to minimize all the tiny vibration that locally affect the system and blurs the spectrum. In fact, we found

indications that some aspect of current work might have been observed in previous STM studies without being noticed or being nailed. It's arguable that spectra resolved in STM is good enough to back the DFT calculation based on resolved geometry in STM studies, exactly by the reason mentioned above. Similar spectra would have different origin and cannot be determined by a single trace of transport behavior. The detailed and decisive transport mechanism and (spin) configuration could only be determined by deep analysis of high resolution transport data which are collected when tuning different knobs. As suggested in the new manuscript (page 9), the illustration of spin 3/2 configuration of MnPc on gold 111 surface determined by previous STS studies is contradictory to the knowledge high magnetic impurity leads to very weak Kondo resonance (under-screened one), while often the case STS studies show very strong resonance peak with Kondo temperature as high as tens of Kelvin. A convincing evidence of underscreened Kondo behavior have never be proved exactly because the current STM technique is still lacking full capability to analysis spectrum. Unfortunately break junction technique in current work usually create a random interface to attach molecules, which is far more complex than the STM configuration with flat metal surface. The random interface and complex local configurations make it impossible to deduce the orbital structure of the molecule inside a device by DFT calculations like what usually do in STM studies. Despite the fact that full orbital structure of molecule can't be deduced, we can easily figure out the spin state associated with different kinds of Kondo effect, and study the tuning of the effect under different kinds of method with sub-meV energy resolution. Different kinds of experimental configurations with specific advantages should be utilized together to fully understand electron transport at molecular scale. We are in pursuing better way to identify molecule structures.

(2) The authors discuss the molecular state by comparing the adsorption of MnPc on flat Au(111) surface. When assuming the similar structure in the device, I am wondering the molecule can be gated. MnPc on Au(111) has been well investigated. Cite the papers such as Phys. Rev. B 84, 125446 (2011) and J. Phys. Chem. C 112, 13650 (2008) and compare the present results with the previous ones.

A2 Gating of single molecule has been widely achieved, although as tiny as 1nm, any local variation might play a significant role. Both works mentioned above are STM studies with DFT calculations. As we pointed out when answering the first question, there is no way we could find gating effect in STM studies. And even there might be some gating effect by local variants, there is no way the STS spectrum can tell the difference. We do realize there are works that carefully put some ions in vicinity of a molecule and study the molecular gating effect while vary the distance between the ion and the molecule (Nature Physics 2015 DOI:10.1038/NPHYS3385). Indeed gating could be achieved even in the STM scheme.

In our last submitting manuscript, we assumed a configuration with molecules adsorption on the side wall of gold electrodes, as the molecules showed a strong unsymmetrical couplings to source and drain electrodes. Recently we discussed our results with Pro. Franck Balestro, who had a lot experience on Pc based single molecule device. According to their results under 3D-vector magnets, except for the configuration we assumed, another configuration with MnPc molecules parallel to the substrate may also exhibit Kondo effect. We can't further prove which configuration works for our devices at this moment. So we leave out this assumption as well as

the comparison to the adsorption of MnPc on Au(111) in this manuscript.

Q(3) The vibrational signatures at 16 and 63 mV are presented as solid evidence of a single molecule device in FIG.S1. The inelastic structures should appear symmetrically in the I-Vsd spectrum. Show more clear spectra like typical cross-sections of 2D color map.

A3 The conductance peaks associate with vibrational excited states of molecules should appear symmetrically on both sides of bias theoretically. However, when the molecule couples to source and drain electrodes unsymmetrically, the excited peak would only appear on one side of bias, as the peak on the other side is too weak to observe, which is very common in single molecule transistor device. In the devices shown in the manuscript, the peak conductance of Kondo resonance is much smaller than the conductance quantum, indicating the significant unsymmetrical coupling in the devices. That's why we only observe the vibrational signature at 16mV and 63mV on one side of bias. This discussion and the cross-sections of 2D color map are added on Fig.S1 in SI.

Reviewer #2 (Remarks to the Author):

The authors present in their work the study of two-stage Kondo effect using a single MnPc single molecule transistor. They perform transport measurements as a function of temperature, gate voltage and magnetic field in the two regions of the Coulomb diamond to discriminate the spin 1/2 to the two-stage Kondo effect. They then perform measurements to show the gate control and the universality of the two stage Kondo effect. The results are important, however, I cannot recommend this work for publication in Nature communications at this stage, as I do not clearly understand the novelty compared to the results obtained by N. Roch et al. Nature Nature 453, 633 (2008) (Quantum phase transition in a single-molecule quantum dot), except the careful study of the evolution of T^* as a function of the singlet-triplet energy. If the authors clearly state the novelty of their work, I will of course change my opinion.

A. Novelty of current paper: Since two stage Kondo effect and later multi-channel multi-level Kondo effect was proposed by Glazemann, Vojta, Hofseter and others, it attracted a lot of research interest hopping to formulate a general picture of Kondo physics and solid fundamental knowledge of exotic quasi particles and processes involving strong correlations. Theoretical framework has not been completed yet for the reason even most successful calculation method, numerical renormalization group, can't treat issues like non-equilibrium process and time dependent process to date. We believe more experimental observation would help and push the thinking. Moreover, the predicted behaviors have yet to be fully checked and validated in the experimental front. Especially, how the unique energy scales T_k and T^* manifest themselves with observables, how they evolve with experimental parameters and how they got linked with basic govern the Many efforts have been put in to study it in quantum dot, carbon nanotube. In term of credit of finding of two stage Kondo effect in molecular devices, we do acknowledge in our manuscript that Nature 2008 paper is the first observation. But the physic and the phenomenon are not clearly addressed. We

could see the spectrum related to two stage Kondo effect changes in the singlet phase regime, while not study of T_K and T^* (the most important factors defining the two stage Kondo effect) as a function of gate were conducted. No information of why Kondo temperatures of two stage Kondo effect changes (or even whether it changed!) with external field or how it changes was obtained till now, although theoretical prediction have been published to be tested since 2002. We believe some of the most important pieces of jigsaw puzzle in multi-level multi-channel are missing, especially on experimental front. The most important innovation of the first part of current (revised version) work is we showed experimentally how Kondo temperature evolve with gate voltage and effective exchange energy. This is equivalent important as gate evolution of Kondo temperature in spin 1/2 Kondo, revealing the key factor of level width and level depth. In two stage Kondo effect, however, the second stage is controlled by coupling of the two levels. We identified and showed how this coupling changes the Kondo temperature, which is the Key information of the Kondo physics. In the second part, we go further to tackle the universal property of Kondo effect in general using the two stage Kondo system we constructed. As we all know, the role of Kondo physics in correlated many-body physics depends on how universal it is. The more universal the phenomena, the wider application ranges. Essentially, the universality indicate the theory, the observation, is Not necessarily limited to the specific system under investigation where detailed nature of interaction raise specific phenomenon only appreciable to that system. The universal behavior could be applied to any physical system that have interaction in similar fashion. Till now, to our knowledge, where is No experimental evidence of universality of Kondo resonance to magnetic field, except a very raw (5 data points, PRL 1997) estimation that could be a quadratic dependence. Here we are talking about all universal study of Kondo physics, not only the two stage Kondo effect. In our case, we experimentally describe the universality in the two stage Kondo regime vs voltage excitation, magnetic excitation and temperature for the first time, in the same device. And we are able to test the recent theoretical prediction when three parameters are tuned. Our achievement not only demonstrate the Kondo effect is more fundamental than thought, but also show the interactions between multi levels and multi channels could be learned by analyzing the coefficient in the quadratic relations. All the message conveyed above are new and in our opinion is original enough for publication.

I also have different comments :

Q1- The authors present in the SI the temperature study of the zero-bias anomaly in region 1. As a saturation of the conductance is clearly observed at low temperature, the authors assume (with a comparison to theory) that a spin 1/2 Kondo effect is observed on the right, with a Kondo temperature of the order of 3.21K. The authors also extracted the g factor performing measurements as a function of the external magnetic field. My comment is : what is the Kondo temperature obtained from the measurement as a function of the magnetic field ? Is it consistent with the T_K obtained as a function of temperature ?

A1. The Kondo temperature is associated with onset critical field for the Zeeman splitting of Kondo resonance. By this method, we estimate the Kondo temperature is 3.32K. Besides, we also

estimated the Kondo temperature from the peak width of Kondo resonance, and the result is 3.78K. All the estimated results are consistent with the $T_{K,1/2}$ obtained from the function of temperature. These discussions are added in Page 2 in SI.

Q2- In Figure 1.c, the authors observe a clear dip in region 2. Measurements as a function of temperature (Figure 2) are consistent with a two-stage Kondo effect. However, in Figure 2.c, it is absolutely not clear that the ground state is the singlet, as the clear signature of the triplet state (3 transitions) should be observed before the critical field, as observed in the work of N. Roch et al., but also in the work of Yu. Bomze et al. Phys. Rev. B 82, 161411(R) (2010) (Two-stage Kondo effect and Kondo-box level spectroscopy in a carbon nanotube). After the critical field, 2 transitions should also be observed. Could the authors comment on why it is not observed ? Or maybe it is just not visible from the figure which is presented, the authors could then apply a second derivative of the conductance to observe this signature.

A2. The magnet study of the splitting peak in Fig.1c is performed at the base temperature (280mK) of our measurement system, which is much higher than the works of N. Roch et al. Nature 453, 633 (2008)(at 35mK) and Yu. Bomze et al. Phys. Rev. B 82, 161411(R) (2010)(at 25mK). The thermal broadening of peaks leads the splitting of the triplet peaks at lower field less pronounced in the dI/dV spectrums. Thanks for the advice, we apply a second derivative of the differential conductance, the splitting signature of triplet as well as the two transitions after the critical field are discernible now(shown in Fig. 2e), revealing a singlet ground state.

This discussing and result are added in Page 4 and Fig. 2e

Q3- In Figure 3.a, the authors present the gate voltage dependence of the singlet-triplet energy splitting. It looks like for more negative gate voltages, it could have been possible to observe the underscreen spin 1 Kondo effect. I presume that the authors did not perform measurements for lower gate voltages to prevent leaking currents or breaking of the Al gate ?

A3. Yes, it should be possible to observe the underscreened spin 1 Kondo effect as well as quantum phase transition in device 1 at a more negative gate voltage. However, as the gate oxide is too thin, the leaking current from gate electrode is significant when the applied gate voltage is larger than 2V(shown in Fig.S3(b) I SI). To reduce the risk for the breakdown of gate oxide, we limit the range of gate voltage from -2V to 2V.

4- Could the authors comment the influence of the magnetic properties of MnPc of the Kondo signatures, as it is not clear for me (I apologize if I missed this point) if transport occurs through the Pc ligand or the Mn atom. My guess is through the Pc ligand as the evolution of the conductance as a function of temperature in region 1 does not fit with NRG calculations for $S=3/2$.

A3. According to the DFT calculation results(J. Phys. Chem. C,112, 13650(2008), J. Phys. Chem. A ,118, 927,(2014)),the magnetic moment (spin) of MnPc molecules mainly attribute to the electrons on the d orbitals of Mn atom. Previous STM studies on MnPc molecules (Phys Rev Lett 109, 147202 (2012), Scientific Reports 3, 1210 (2013)) also show that the Kondo effect appears on the center of the molecule, i.e. the electron transport mainly through Mn atom but not the Pc ligand. Since the referee is very familiar with NRG result of $S=3/2$ Kondo resonance, we would like

to point out none of above experimental efforts have successfully shown the $S=3/2$ characteristics in temperature dependent measurement or bias dependent measurement. Exact reason of this is unknown to our limited knowledge. In our case, we could believe our observation is consistent with above STM studies that Mn ion is most likely responsible for the observed two stage Kondo observation. The reduced spin from $3/2$ to $1/2$ maybe arise from the ligand field change induced by the complex interface between molecule and gold electrode in our devices, with random atom arrangements on the surface. Detailed discussion is added on Page 10.

REVIEWERS' COMMENTS

Reviewer #1 (Remarks to the Author):

The energy diagram and spin of MnPc are still elusive, but the discussion is strengthened by citing the previous experimental and theoretical studies reasonably. In addition, the detailed analysis on the scaling has been added. These revisions make the manuscript much better. The systematic investigations of two-stage Kondo effect as functions of temperature, magnetic field and bias voltage provide important information to promote our understanding of Kondo physics, strong correlation and non-equilibrium transport in molecular devices. Thus, I recommend the manuscript for publication in nature communications.

Reviewer #2 (Remarks to the Author):

The authors answered adequately to my concerns as well as to the comments of the other referee. I appreciate the modifications they introduced in the main text in order to gain clarity. Thus, I recommend this work (without any more modifications) for publication in Nature Communications.